# Association of Dengue Virus Serotypes 1&2 with Severe Dengue Having Deletions in Their 3′Untranslated Regions (3′UTRs)

**DOI:** 10.3390/microorganisms11030666

**Published:** 2023-03-06

**Authors:** Deepti Maisnam, Arcy Billoria, V. S. V Prasad, Musturi Venkataramana

**Affiliations:** 1Department of Biotechnology and Bioinformatics, School of Life Sciences, University of Hyderabad, Hyderabad 500046, Telangana, India; 2Lotus Children Hospital, Lakdikapul, Hyderabad 500004, Telangana, India

**Keywords:** dengue virus, severe dengue, untranslated regions (UTRs), flaviviruses, DHF, DSS

## Abstract

Dengue virus infections are recorded as hyper-endemic in many countries, including India. Research pertaining to the reasons for frequent outbreaks and severe dengue is ongoing. Hyderabad city, India, has been recorded as a ‘hotspot’ for dengue virus infections. Dengue virus strains circulating over the past few years in Hyderabad city have been characterized at the molecular level to analyze the serotype/genotypes; 3′UTRs were further amplified and sequenced. The disease severity in patients infected with dengue virus strains with complete and 3′UTR deletion mutants was analyzed. Genotype I of the serotype 1 replaced genotype III, which has been circulating over the past few years in this region. Coincidentally, the number of dengue virus infections significantly increased in this region during the study period. Nucleotide sequence analysis suggested twenty-two and eight nucleotide deletions in the 3′UTR of DENV-1. The eight nucleotide deletions observed in the case of DENV-1 3′UTR were the first reported in this instance. A 50 nucleotide deletion was identified in the case of the serotype DENV-2. Importantly, these deletion mutants were found to cause severe dengue, even though they were found to be replication incompetent. This study emphasized the role of dengue virus 3′UTRs on severe dengue and emerging outbreaks.

## 1. Introduction

The dengue virus is considered a global threat to mankind due to its augmenting rate of infection in tropical and sub-tropical regions [1]. It constitutes symptoms ranging from simple febrile illness to severe complications leading to hemorrhage and shock. The WHO classified dengue as dengue with and without warning signs, and severe dengue [2]. Around 390 million annual dengue cases were recorded by the WHO and the rate has increased by thirty-fold over the last five decades. Nearly 130 countries are now endemic to dengue, putting 3.9 billion people at risk, of which 70% burden is accounted for by Asian countries [3]. Dengue virus belongs to the *Flaviviridae* family, which is mostly transmitted by *Aedes aegypti* and, to some extent, by *Aedes albopictus* mosquitoes. The viral genome consists of a single stranded positive sense RNA, with an approximate size of 11 kb coding for a single open reading frame (ORF). This ORF codes for three structural (Capsid, pre-Membrane and Envelope) and seven non-structural proteins (NS1, NS2A and 2B, NS3, NS4A and 4B and NS5). Two untranslated regions (UTRs) flank the viral genome, one on each 5′ and 3′ ends. These UTRs play a pivotal role in the virus lifecycle, being involved in replication and translation by forming stem loop structures. Reports have indicated that diversity in these genes is also of major concern with regard to virus evolution [4]. 

The dengue virus circulates as four different serotypes (DENV1–4), each categorized based on their antigenic differences. These serotypes are further sub-categorized into various genotypes which differ by 3 and 6% at the amino acid and nucleotide levels, respectively. DENV-1 has six genotypes (GI-VI), DENV-2 also has six genotypes (Asian American, Asian I, Asian II, Cosmopolitan, American and Sylvatic), DENV-3 has five genotypes (GI-V) and DENV-4 has four genotypes (GI-III including one sylvatic strain) [5]. Earlier epidemics were caused by single serotypes, but this later changed to multiple serotypes leading to co-infections [6,7]. In dengue-virus-infected regions, DENV-2 infections were higher, the rate of co-infection was estimated to be 47.7% and high mortality was also observed [8]. However, a serotype shift has been observed in recent years, and the worst situations have been modeled [9]. It appears that virulent genotypes displace the non-virulent strains [10]. Inter-serotype recombination has been reported, which leads to evolving new strains [11]. Hence, the identification of serotypes and circulating genotypes is one of the factors that is being considered in order to initiate the prevention and control of the dengue virus. Hyderabad, the capital city of Telangana state, India, has been recorded as a ‘hotspot’ for dengue virus infections [12]. In this study, dengue-virus-infected clinical samples were collected from this geographical region from 2017 to 2020 and were characterized. An attempt was made to elucidate the dengue virus dynamics and disease severity by analyzing the sequences of 3′UTRs. 

## 2. Materials and Methods

### 2.1. Clinical Samples

In the present study, a total 120 samples from suspected cases of dengue were collected from 2017 to 2020 from Lotus Children’s Hospital, Lakdikapul Road, Hyderabad. Informed consent was obtained from the parents or guardians of the children. The samples with Dengue NS1 antigen (Denguecheck) and IgM/IgG ELISA (Mac-elisa IgM/IgG Microlisa kit) positive were included in the study and the samples negative for the above tests or positive for the above tests, along with other diseases, were excluded. The collected serum samples were stored at −80 °C until use. Segregation of the samples into different categories of dengue infection was carried out based on WHO guidelines 2009 [2]. 

### 2.2. Serotyping 

Viral RNA was extracted from 140 uL of the serum sample using a QiAamp viral RNA mini kit (Qiagen, Germany) according to the manufacturer’s instructions. Using 5 µL of the above isolated RNA as a template, cDNA was synthesized with AMV Reverse transcriptase (NEB) at 45 °C for 60 min using 10 pM Capsid pre-Membrane (CpRM) reverse primer. The above cDNA was further amplified by both forward and reverse primers of the CpRM region. Using the amplified CpRM product as a template, serotyping was performed using serotype-specific primers as described earlier [6]. The amplified products were resolved by 1.5% agarose gel electrophoresis, visualized and then recorded. The obtained sizes of the products for DENV-1, DENV-2, DENV-3 and DENV-4 were 482, 119, 292 and 390 bp, respectively. A Pie diagram was prepared with the number of different serotypes individually identified or in co-infections. 

### 2.3. Sequencing and Phylogenetic Analyses

The CpRM region of DENV-1 (5 samples), DENV-2 (1 sample) and DENV-3 (1 sample) were amplified and the amplicons were gel purified (kit from Takara, Japan). The extracted product was cloned in a pJET 1.2/blunt end vector (CloneJet PCR cloning kit, Thermo Scientific) and commercially sequenced (Eurofins, India). Sequencing was repeated three times and consensus sequences were submitted to the gene bank database.

The accession numbers were obtained: OM350002, OM350003, OM350004, OM350005, OM350006, OM982618 and OM924030. The sequences were aligned using Clustal W multiple alignment along with existing dengue sequences from the NCBI database. Based on the general time reversible model with gamma distribution and invariant sites keeping bootstrap values of 1000 replicates, the phylogenetic tree was obtained using Molecular Evolutionary Genetics Analysis X (MEGA X) software (Version 10.1.5). Sequencing of the DENV-4 amplified product was unsuccessful.

### 2.4. Epidemiological Data of Dengue Cases in Telangana

In order to obtain the status of the dengue cases in the state of Telangana during the study period, we analyzed the epidemiological data of the state available on the National Vector Borne Disease Control Program (NVBDCP) website (https://nvbdcp.gov.in; accessed on 22 December 2022). The number of cases of dengue infections was considered from 2015 to 2020. Based on this data, a line graph was plotted with the year on the *x*-axis and the number of cases on the *y*-axis. 

### 2.5. Analysis of 3′ Untranslated Regions (3′UTRs)

From the serotyped DENV-1-positive samples, the 3′UTR was amplified by RT-PCR using the reverse primer of 3′UTR, and then was further amplified by using both the forward and reverse primers. The amplified products were cloned into pJET 1.2/blunt end vector and sequenced. The NCBI accession numbers for the submitted sequences were MZ461927, MZ461928, MZ461929, MZ461930, MZ461931, OM572555, OM572556, OM572557, OM572558, and OM572559. Using Clustal W multiple alignment of BioEdit software (7.2.5), 3′UTR sequences of other DENV-1 strains from the NCBI database, including earlier sequences from the laboratory, were aligned with the sequences obtained in the present study. In a similar manner, sequences of DENV-2 3′UTRs were aligned. 

### 2.6. Analysis of Replication Efficiency of 3′UTR Mutants

#### 2.6.1. Quantitative Real-Time PCR (qPCR)

The Vero E6 cells were obtained from the National Centre for Cell Science (NCCS), Pune, India, and, upon receipt, the cells were maintained in Dulbecco’s Modified Eagle Medium (Gibco) with 10% Foetal Bovine Serum (Gibco) and 1% of Antibiotic-Antimycotic (Thermo Fisher) until its full growth potential. The virus cultures (full length and ∆22 3′UTRs), confirmation, and quantification were performed as described earlier [13]. 

In order to develop the standard curve of the qPCR, the cDNA encoding the 5′UTR of the dengue virus genome was amplified using a specific primer set [14]. Then, the amplicons were ligated to the pJET 1.2/Blunt end vector. This construct, in the order of 10^3^ to 10^10^ copies, were subjected to qPCR and used for Standard curve preparation. 

Viral RNA was isolated using the Qiagen viral RNA isolation kit from infected and uninfected viral supernatants. The obtained RNA was also quantified and, from 1 µg of the viral RNA, cDNA was prepared using the PrimeScript cDNA synthesis kit (Takara). This DNA was used in the real-time PCR reaction along with 0.1µM of both the primers using USB VeriQuest SYBR Green qPCRMaster Mix (2×) (Affymetrix). The reaction conditions were as follows: 50 °C for 2 min, 95 °C for 10 min, and 45 cycles of 95 ℃ for 15 s, annealing at 60 °C for 30 s, followed by extension at 72 °C for 30 s. A standard curve was generated using the C_t_ values of the 5′UTR cDNA clone, in which 10^3^ to 10^10^ copies of plasmid were used. The respective average C_t_ values were plotted on the Y-axis with log plasmid concentrations on the X-axis. From these values, a linear regression equation was obtained and, based on the equation, the copy numbers of both full-length and mutant 3′UTRs were obtained. Using the copy numbers, bar graphs were generated with standard deviation as error bars.

#### 2.6.2. Structural Analysis of 3′UTRs 

DENV-1 UTR RNA structures were prepared using the UNAFold Web Server (http://www.unafold.org/mfold/applications/rna-folding-form.php; accessed on 18 June 2022) with all the default folding parameters and the folding was predicted at 37 °C at the optimum energy levels. The structures with the lowest free energy were considered for further analysis. The strains used include the full-length and deleted strains of the present study (Accession No. MG560143 and MG560144). A similar method was employed for developing the DENV-2 3′UTR RNA structures.

## 3. Results

### 3.1. Patient Samples

Hyderabad, the capital city of Telangana state, in southern India, is the health hub for more than 10 million people. Reports of several viral infections have been recorded from this region. The suspected cases of dengue included in this study had symptoms like fever, headache, fatigue, and joint pain, along with severe symptoms such as thrombocytopenia, abdominal pain, pleural effusion, and liver enlargement, etc. The platelet counts were in the range of 30,000–180,000/mL. 

### 3.2. Analysis of Serotype

Among the total 120 samples used in this study, 51 samples were successfully subjected to serotyping (Figure 1a). The data suggested that DENV-1 was detected in thirty samples (59%), DENV-2 in four samples (8%), DENV-3 in eight samples (15.6%), and DENV-4 in one sample (2%) (Figure 1b). The remaining samples were co-infected (19%) with different serotype combinations like DENV-3&4 (3.92%), DENV-2&4 (1.96%), DENV-1&4 (5.88%), DENV-2&3 (5.88%), DENV-1&3 (1.96%), and DENV-1&2 (1.96%). DENV-1 was found to be the dominating serotype in the present study, whereas the co-infection rate was found to be less in comparison to the previous report from the state (Figure 1b). 

### 3.3. Identification of Genotype

CpRM sequences from a total of seven samples (five samples of DENV-1, one each from DENV-2 and DENV-3) were cloned and sequenced for phylogenetic analysis. The phylogenetic analysis suggested that all of the DENV-1 samples were found to be of Genotype I (Asian genotype) (Figure 2). This genotype was found to replace genotype III, which was predominantly circulated in this geographical region. Further, the data indicated that this genotype was closely clustered with the Kerala strain, which was circulated in 2013, and the Sri Lanka strain, which was circulated in 2012 (Figure 2). The cosmopolitan genotype of DENV-2 and Genotype III of DENV-3 were found to be the same as in the region earlier (Appendix A). 

### 3.4. Analysis of Dengue Cases from 2015 to 2020

Since there is genotype displacement (GIII to GI) in the case of dengue virus serotype 1, we analyzed the dengue cases for the past 5 years in the Telangana state published by the National Vector Borne Disease Control Program (NVBDCP). The data indicated that the number of dengue cases dramatically increased during 2018–2019 (Figure 3). Hence, it was concluded that the change in genotype could be one of the reasons for the rise in the number of dengue cases in this geographical region. 

### 3.5. 3′UTR Sequence Analysis

The reports indicate that the virus genome UTRs contribute to disease severity and emerging outbreaks [4]. Hence, we carried out sequence analysis of the 3′UTRs of the DENV-1 strains characterized in the present study. The 3′UTR sequences of DENV-1 from NCBI, sequences from our earlier study, and the strains of the present study were used for the analysis. The data showed few strains with full-length UTRs, whereas the other strains had twenty-two and eight nucleotide deletions (Figure 4). While the twenty-two deletions were also reported from other regions, the eight nucleotide deletions were not reported until now. Both the twenty-two and eight nucleotide deletions were observed in the hypervariable region of the 3′UTR. The sequence alignment of DENV-2 also showed a deletion of 50 nt in the stem-loop region (Accession No. MG560144), whereas the others had full-length sequences (Accession No. MG560143) (Appendix A). In the case of DENV-2, the deletions were found towards the end of the 3′UTR. 

### 3.6. Viral Copy Number of Full-Length and Mutant 3′UTR Samples

The viral copy numbers of both the full-length and deleted strains were determined by qPCR to measure replication efficiency. We found that the full-length strain had approximately 8125 copies of RNA, while for the deleted strain, it was 4554 copies (Figure 5). Based on this data, it was concluded that there was approximately a 1.5-fold lesser replication rate in the deletion mutants compared to the strains with full-length 3′UTRs. Interestingly, the clinical characteristics of patients infected with strains with 3′UTR deletions were found to be also associated with severe dengue (Table 1). 

### 3.7. RNA Structure Analysis of the 3′UTRs

Since the function of the flavivirus 3′UTR depends on both the primary sequence and secondary structures, we performed in silico 3′UTR RNA structural analysis of the study strains, i.e., full-length and deleted sequences. In the full-length sequence (Acc. no. MZ461930), there was an extra bulge loop (Figure 6a) that was formed by the sequences which were deleted in the mutant strains (Figure 6b,c). This region covers the hypervariable region. However, the structures of all three sequences showed similar stem-loop structures in the conserved regions. This is in accordance with the earlier study (P23086) [15]. A similar structural analysis of DENV-2 showed the presence of an elongated 3′stem loop (3′SL) in the full-length strains in comparison to the deleted strains. There was a remarkable difference in the length of the stem-loop formed by nucleotides from 400 to 450, but these nucleotides were absent in the mutant strains (Appendix A). It will be interesting to see the effect of this deletion in terms of virus replication efficiency and virulence. 

## 4. Discussion

Several of the dengue-virus-infected countries have been recorded as hyper-endemic for dengue infections circulated by all four serotypes. The search for the reason behind severe dengue and the markers for future outbreaks is ongoing. Many outbreaks have been reported from all over the world with either the same or different serotypes causing mild to severe infections [16]. Until 2010, dengue outbreaks were dominated by serotype 2 in India. However, DENV-1 dengue outbreaks have increased, especially over the last 10 years [8]. It has been reported that multiple lineages of GIII of DENV-1 have circulated in India from 1956 to 2007 [17]. Genotype III of this serotype was dominant during the years 2012–2013, but GI showed predominance during 2014–2015 in Karnataka state, southern India [18]. In this direction, GI displaced GIII from the rest of the country for the past 3–4 years [17,18,19]. Southern states of India, like Kerala, Karnataka, Tamil Nadu, Andhra Pradesh, and Telangana, have reported all four serotypes [6,20]. However, recently, different genotypes have been reported in addition to the existing genotypes. They are GI of DENV-1, GIV of DENV-2, and GIV of DENV-3 [19]. This poses a major concern, as changes in genotypes may cause more dengue outbreaks and an increase in severity [21]. 

In the present study, we conducted a genetic characterization of DENV-1, DENV-2, DENV-3, and DENV-4 circulating in the city of Hyderabad, Telangana, India. The data suggested that, out of the total 51 serotyped samples, the DENV-1 serotype dominated with 59% (Figure 1b). However, reports suggested the circulation of all four serotypes in this geographical region in nearly equal proportion a few years ago [6,20]. This change led to further analysis of the genotypes of DENV-1. The data showed the replacement of GIII by GI. This genotype has been reported in other southern states, such as Kerala, Karnataka, and Tamil Nadu [18,22]. All of the five sequences subjected to genotype analysis showed close relatedness to the Kerala (Accession no. KJ755855) and Sri Lanka strains (Accession no. KT445959) (Figure 2). This genotype represents strains from Southeast Asia, China, and the Middle East [23]. It is predicted that genotype 1 of DENV-1 has been introduced from Thailand through Sri Lanka, and subsequently, to Tamil Nadu, which is located adjacent to Sri Lanka. On the other hand, the phylogenetic analysis of DENV-2 and DENV-3 showed the circulation of GIV and GIII, respectively, as reported in 2014. The GIV of DENV-2 has a wide distribution range, as its presence has been detected in areas such as the Middle East, Africa, Australia, and the Indian subcontinent. These genotypes have been associated with severe dengue cases and increased co-infection rates. Such continued circulation of the same genotype can also represent a chance to continuously pose a threat to the public.

The introduction of a new genotype has the ability to cause a greater number of dengue infections in the population, either as single infections or co-infections. In this direction, since the genotype change was observed in the case of DENV-1 in this geographical region, we have analyzed the number of cases from 2015 to 2020 (Figure 3). The data suggested a steep rise in the number of cases during 2018–2019, which may be due to the change in the DENV-1 genotype from GIII to GI. The WHO has also reported that the global dengue infection rate was highest in the year 2019, with an estimated 5.2 million people infected. It appears that the number of dengue virus infections may stabilize or even come down if no change takes place in the circulating serotype or the genotypes. Hence, continuous monitoring of serotypes/genotypes may facilitate the prediction of dengue virus infections in subsequent years.

The untranslated regions (UTRs) present at both ends of the genome are known to play a vital role in genome replication/translation. The 5′UTR is nearly ~100 nts in length and is highly conserved. Whereas the 3′UTR is nearly ~450 nts in length and is divided into highly variable, moderately conserved, and conserved regions [24]. The 3′UTR is known to play a major role in virus multiplication and disease pathogenesis compared to the 5′UTR [25]. The possibility of increased transmission and epidemic situations has been reported if any minor genetic changes occur in flavivirus RNA [26]. Furthermore, it has been reported that genetic variation in 3′UTR contributes to epidemic emergence [27]. Several reports have suggested that the insertions/deletions in 3′UTR have recently contributed to disease pathogenicity [4]. It was also suggested that insertions were helpful in replication, in contrast to deletions. Strains with insertions have been found to replicate better, be more virulent, and cause cell death. Whereas strains with deletions do not cause cell death [28]. Studies have also suggested that the deletions are dispensable [29]. Considering the above literature, we analyzed nucleotide sequences of the 3′UTRs of the DENV-1 strains of the present study, along with the DENV-1 and DENV-2 strains previously characterized in our laboratory. The data indicated DENV-1 and 2 strains with deletions in their 3′UTRs. There were twenty-two and eight nucleotide deletions in the case of DENV-1, and fifty nucleotide deletions in the case of DENV-2 (Figure 4 and Appendix A). The twenty-two nucleotide deletions were observed in both GIII and GI of DENV-1. Hence, the 3′UTR of each serotype may be specific. This observation is supported by earlier studies which suggested that strains with interchanged 3′UTRs show attenuation [30]. The twenty-two nucleotide deletions were present in the hypervariable region of the 3′UTRs of both genotypes (GIII and GI) of DENV-1, whereas the fifty nucleotide deletions of the DENV-2 serotype were located towards the end of the 3′UTR in the highly conserved stem-loop (SL) structure. The location of the deletions observed in each serotype also appears to be specific, irrespective of the genotype. Hence, it is speculated that the observed deletions are part of dengue virus evolution. The above observation is supported by the fact that the 3′UTR, along with the E gene, are recognized as ‘hotspots’ for mutation and evolution [10]. The eight nucleotide deletions observed in the case of DENV-1 were not reported earlier, and hence, required additional studies to reveal the consequences.

The literature suggests that strains with deletions were replication-incompetent, and hence, proposed to use as vaccine candidates [30]. Although the strains in the present study were found to be replication-incompetent (Figure 5), some of these strains have been associated with severe dengue (DHF and DSS) (Table 1). In addition, reports have also indicated that the dengue virus outbreaks were dominated by DENV-2 during the period 2010–2015, during which the 3′UTR deletions were observed. After 2010, the outbreaks were dominated by DENV-1, during which the 3′UTR deletions were noticed in this serotype [31]. Hence, it appears that dengue virus strains also evolve as deletion mutants, as exemplified above, to cause severe dengue or new outbreaks by discarding the part of the genome which is dispensable. Another relevant aspect is the role of miRNAs and cytokines in UTRs. Virus infections induce cellular cytokine and miRNA production. miRNAs are known to bind to the virus 3′UTRs and are also known to downregulate the host’s anti-viral response through interferon targeting. Dengue infections induce an array of cytokines based on the severity of the disease [32,33]. Reports have indicated that anti-viral responses in dengue virus infections will be triggered by miRNAs [34,35,36]. Hence, it appears that there is an interplay between virus infection, cytokine induction, miRNA production, 3′UTRs, and severe disease/clearance [37,38,39,40]. Considering the above interplay, the mechanisms of severe disease caused by strains with deletions needs to be further deciphered. 

## Figures and Tables

**Figure 1 microorganisms-11-00666-f001:**
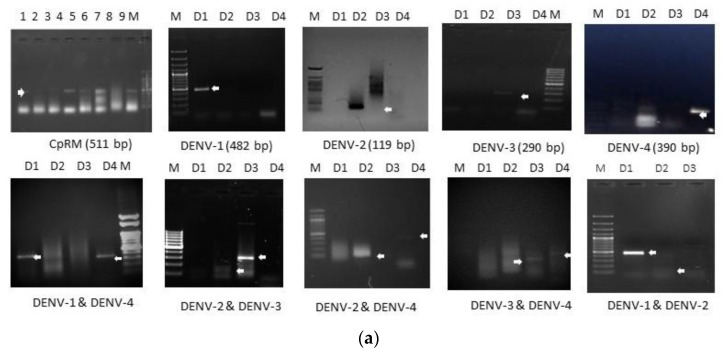
Serotype analysis in the clinical samples collected during the years 2017–2020. (**a**) Agarose gel electrophoresis images of the serotypes after RT-PCR and Nested PCR amplification with their specific sizes. (**b**) Pie chart representing the percentage of samples identified for different serotypes and co-infections.

**Figure 2 microorganisms-11-00666-f002:**
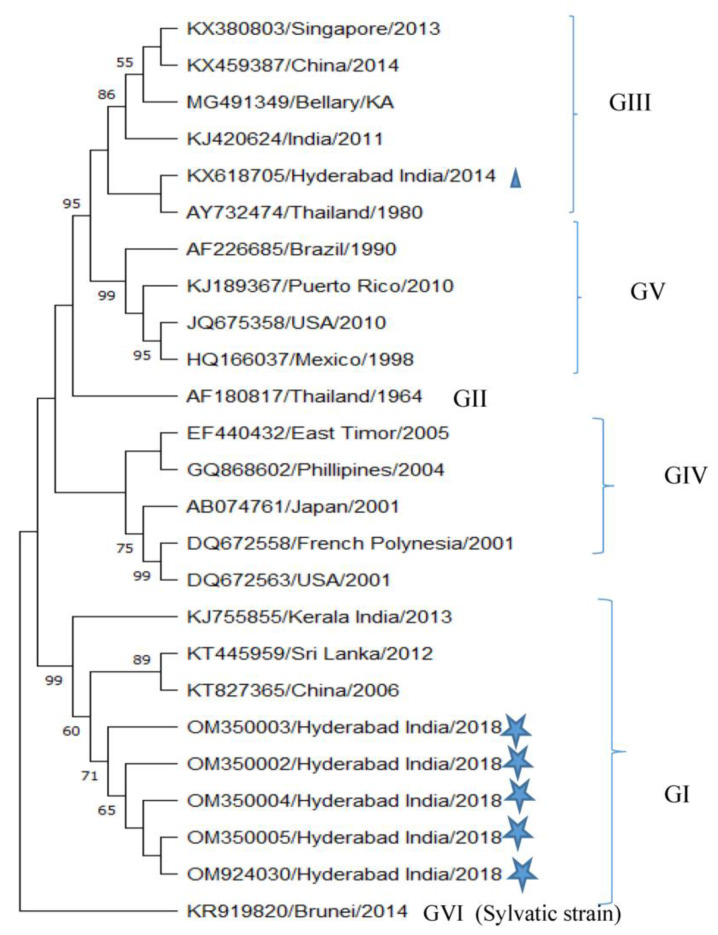
Phylogenetic tree of DENV-1 based on CpRM sequence generated by general time reversible model as described in the methods. The study sequences are marked by the 

 symbol, which are clustered in GI (Asian genotype). Previous GIII (sylvatic strain) is also indicated by the 

 symbol. The sequences are indicated by their accession number, country, and year of collection.

**Figure 3 microorganisms-11-00666-f003:**
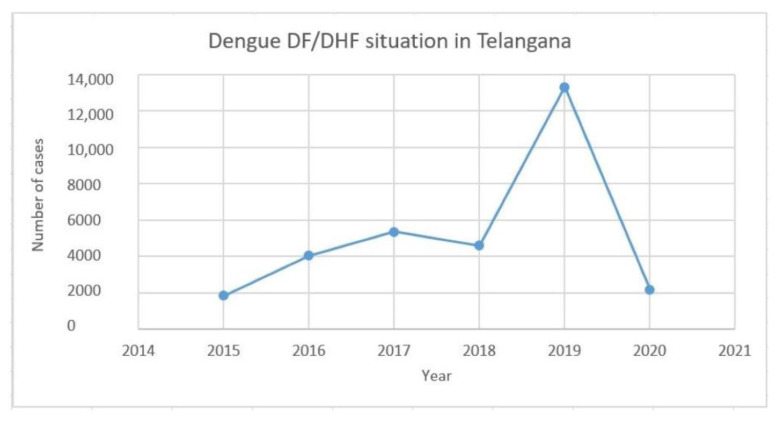
Epidemiological trend of dengue cases in the state of Telangana from 2015 to 2020 based on data available from the National Vector Borne Disease Control Program (NVBDCP).

**Figure 4 microorganisms-11-00666-f004:**
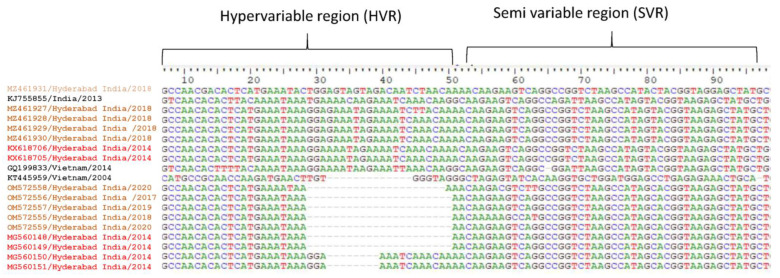
Comparative nucleotide sequence alignment using DENV-1 3′UTR sequences obtained from the NCBI database (black) and the sequences identified from the present study (orange), along with previous sequences from the laboratory (red).

**Figure 5 microorganisms-11-00666-f005:**
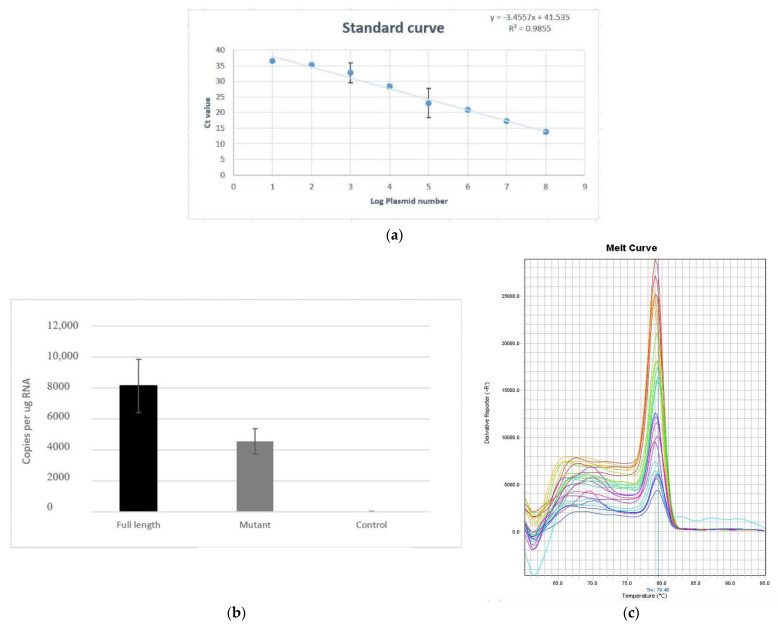
Quantitative real-time PCR (qPCR) data obtained from viral RNA as a template, indicating (**a**) Standard curve generated from the data obtained with R^2^ of 0.9855. (**b**) Bar diagram indicating the RNA copy number per µg of RNA from the strains with full-length 3′UTR, deletion mutant, and the negative control. (**c**) The melt curve.

**Figure 6 microorganisms-11-00666-f006:**
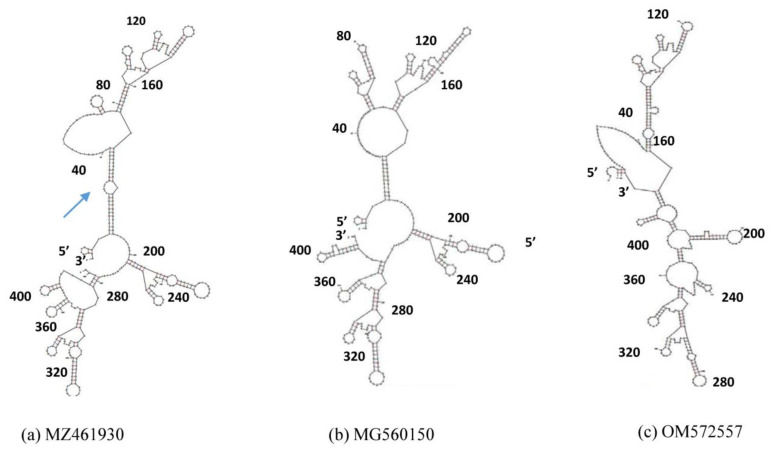
Predicted 3′UTR RNA secondary structure developed using the UNAfold web server for the three strains with accession numbers (**a**) MZ461930 (full sequence), (**b**) MG560150 (∆8 nts), and (**c**) OM572557 (∆22 nts).

**Table 1 microorganisms-11-00666-t001:** Clinical characteristics of DENV-1 and 2 samples with full length and deletions in 3′UTRs.

Serotype	Accession No.	Type of Dengue	No. of Deleted Nucleotides	Age/Sex	Thrombocytopenia	Pleural Effusion	Abdominal Pain	Hematomegaly	NS1/IgM/IgG	AST/ALT
DENV-1	KX618705	DF	Full length	8 Y/M	Yes	No	No	No	+/−/−	Mild
DENV-1	KX618706	DHF	Full length	12 Y/F	Yes	No	Yes	Mild	+/−/−	Normal
^a^ DENV-1	MG560149	DHF	22	8 M/M	Yes	Yes	Yes	Yes	+/−/−	Normal
^a^ DENV-1	OM572555	Dengue with warning signs	22	2 Y/M	No	No	No	No	+/+/−	Normal
^a^ DENV-1	OM572558	Severe dengue with myocarditis	22	4 Y/F	Mild	Yes	Yes	Yes	+/+/−	Mild
DENV-1	MG560150	DF	8	3 Y/F	No	No	No	Yes	+/−/−	Normal
DENV-1	MG560151	DF	8	11 Y/M	Yes	No	Yes	No	+/−/−	Elevated
^a^ DENV-2	MG560144	Severe dengue	50	13 Y/M	Yes	Mild	Yes	Yes	+/−/−	Elevated
DENV-2	MG560143	Severe dengue with Septic shock	Full length	7 Y/F	Yes	Yes	Yes	Yes	+/+/−	Elevated

^a^ Denotes the deletion mutants causing the severe dengue.

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
