# Peer review of "Association of Dengue Virus Serotypes 1&2 with Severe Dengue Having Deletions in Their 3′Untranslated Regions (3′UTRs)"

_microorganisms, 2023, doi:10.3390/microorganisms11030666_

Round 1

Reviewer 1 Report

Dear authors, 

It a good paper 

Thanks

Reviewer 2 Report

The article 'Association of dengue virus serotypes 1&2 with severe dengue having deletions in their 3’untranslated regions (3’UTRs)' by Deepti Maisnam et al. describes the co-relation between deletions in the UTR region and dengue severity. My comments are as follows

1.     The article is well written and provides good introduction and background.

2.     The figures in the article must be improved. In figure 1a, the gel images do not describe what the authors are trying to show. The first gel image is not labelled. In the third gel image the samples are in a different order. In figure 1b, the color scheme used makes it difficult to separate coinfection and DENV-1 cases. In figure 2, the figure legends are not clear. I am not sure what symbol is marking the sequences from this study.  

3.     Plotting data from National Vector Borne disease control program (NVBDC) can’t be included as a separate figure. Also, the authors provide a conclusion that increased number of cases might be due to changes in the genotype without providing any context.

4.     The viral copy number alone can’t be used to provide replication efficacy. The state of the disease progression, the day when the samples were collected after the onset of symptoms and also individual immune response all have an effect on viral copy number. The viral replication efficiency can only be confirmed in the lab with in-vitro studies.

5.     The table 1, only four cases of either DHF or severe Dengue can be attributed to mutation in the UTR region. Overall, the conclusion that mutations in the UTR are associated with severe dengue is very weak and the current data, sample size and absence of any statistical analysis is not enough to establish this association.

Reviewer 3 Report

They did a long-term nucleotide sequence analysis and identified fragment deletions in 3’UTR of some strains of Dengue virus that may related to enhancement of virus pathogenicity. This may be an important discovery with clinical relevance. Generally, it is a good study and the manuscript is well written. However, I have a few concerns.

Although the findings may be of importance, the study is incomplete which may affect the reliability and significance of the conclusions. The sequence change in the 3’UTR of DENV genome may affect viral replication or viral protein translation. They realized these, but they just detected the viral replication and found it was incompetent. How the deletions in 3’UTR result in severity of DENV infection should be further studied and clarified. The translation efficiency of Dengue virus with deletions or not can be detected, or the activities of different 3’UTRs can be evaluated by constructing into some model plasmid such as luciferase reporter system (Protein Cell 2013, 4(2): 130–141; Journal of Biological Chemistry. 2013; 288(29):20927–41). If the deletions do not affect the 3’UTR activities for replication, translation or other process, the conclusion of the study will remain undetermined.

Besides, there are some problems of nonstandard writing such as the name of Dengue viruses. Den1, Den2, Den3 and Den 1, Den 2, Den 3 (with space) were all used in the text, but DENV-1 and DENV-2 were used in the table. Please use the standard virus name and keep unified.

The 3’UTRalso written as 3’ UTR”(with spaceand 3’. UTR. Please unify them.

There are punctuation symbols at the end of paragraph headings such as “1. Introduction:”, “3. Results:”, “3.4. Analysis of dengue cases from 2015 to 2020:”, “3.5 3’. UTR sequence analysis-”, “4. Discussion:”, and other paragraph headings are not punctuated.  

Reviewer 4 Report

1. In Table 1, Figure 5b and Abstract sections:

The authors emphasized that 3’UTR deletion of Den 1 and Den 2 mutants were found to cause severe dengue, but the patients infected with full length 3’UTR dengue virus also developed severe DHF (KX618706, MG560142). On the contrary, patients infected with deleted Den 1 only show mild DF (MG560150, MG560151). What the reason of these deletion mutants cause severe dengue are found to be replication incompetent?

Minor comments:

1. In Figure 4, Abstract sections:  The GenBank number OM572557, OM572556, OM572555, OM572559, et al., their deletion base is 22 not 21; also MG560150 and MG560151, their deletion base is 8 not 7. The MG560150 and MG560151 deposition time is 2014, not in this study period 2015-2020.

2. Other dengue 1, 2  genome region mutations might be  also important in viral pathogenesis, but in this study only 3' UTR sequences are showed.

Round 2

Reviewer 4 Report

The authors had corrected the mentioned suggestions.